# A TRAINABLE MANIFOLD FOR ACCURATE APPROXIMATION WITH ReLU NETWORKS

## ABSTRACT

We present a novel technique for exercising greater control of the weights of ReLU activated neural networks to produce more accurate function approximations. Many theoretical works encode complex operations into ReLU networks using smaller base components. In these works, a common base component is a constant width approximation to $x^2$, which has exponentially decaying error with respect to depth. We extend this block to represent a greater range of convex one-dimensional functions. We derive a manifold of weights such that the output of these new networks utilizes exponentially many piecewise-linear segments. This manifold guides their training process to overcome drawbacks associated with random initialization and unassisted gradient descent. We train these networks to approximate functions which do not necessarily lie on the manifold, showing a significant reduction of error values over conventional approaches.

## 1 INTRODUCTION

To approximate specific classes of functions, a deep ReLU network may exponentially outperform a shallow one. They can do so by using the activation function in a manner that generates an exponential number of linear regions with respect to depth. However, a complete theoretical understanding of how this may be done in general is lacking, which may hinder approximation power. For example, Frankle & Carbin (2019) show that up to 90% of neurons may be safely discarded from a randomly initialized network without affecting accuracy. In a similar vein, Hanin & Rolnick (2019) bound the usage of ReLU in a randomly initialized network. They prove that the expected number of constant slope regions does not scale exponentially with depth; rather it is independent of the configuration of the neurons, and gradient descent was not observed to significantly improve over the bound. Our work aims to eliminate these inefficiencies in a one-dimensional setting. We reason about a generalization of an exiting theoretical technique, using our results to impose an efficient weight structure during training, reducing the reliance on randomness. While the functions we learn are simple, the key significance of our work is that it demonstrates how to incorporate theoretical results about function representation to produce superior training regimes. Our central contribution is the following two part training procedure:

1. First, we initialize the networks onto a manifold of weights which ensures that the output will use an exponential number of line segments in its approximation. The network is then reparameterized and trained in terms of the manifold parameters.
2. The underlying weights of the network are freed and trained directly by gradient descent.

Later in section 4.4, we discuss how the number of linear segments used in an approximation is not a property gradient descent can directly optimize. New "bends" are difficult to discover without assistance. Our procedure not only initializes networks to efficiently leverage depth from the outset, but it ensures they continue to do so throughout the training process. At any point in the first stage manifold, our approximations will use $2^{<depth>}$ line segments. This process prevents gradient descent from taking short-sighted decisions which may sacrifice approximation power for near-term accuracy gains. This reliably produces error values orders of magnitude lower than a traditionally implemented network of equal size.

## 2 BACKGROUND

### 2.1 RELATED WORK IN APPROXIMATION

Infinitely wide neural networks are known to be universal function approximators, even with only one hidden layer (Hornik et al., 1989), (Cybenko, 1989). Infinitely deep networks of fixed width are universal approximators as well (Lu et al., 2017), (Hanin, 2019). In finite cases, the trade-off between width and depth is often studied. There are functions that can be represented with a sub-exponential number of neurons in a deep architecture which require an exponential number of neurons in a wide and shallow architecture. For example, Telgarsky (2015) shows that deep neural networks with ReLU activations on a one-dimensional input are able to generate triangle waves with an exponential number of linear segments (shown in figure 1 as T(x)). The way this network functions is as follows: Each layer takes a one dimensional input on $[0, 1]$, and outputs a one dimensional signal also on $[0, 1]$. The function they produce in isolation is a single symmetric triangle. Together in a network, each layer feeds its output to the next, performing function

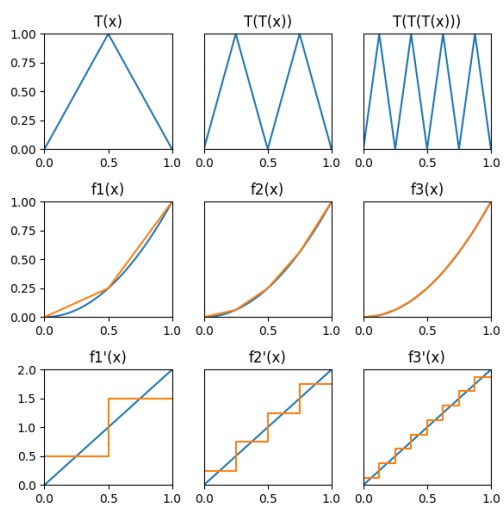

Figure 1: composed triangle waves, approximation to $x^2$, and the derivative

composition. Since each layer converts converts lines from 0 to 1 into triangles, it doubles the number of linear segments in its input signal, exponentially scaling with depth. Our networks will employ a similar strategy to this one, we will allow for the peaks of the individual layers to occur anywhere on $(0, 1)$ rather than fixing them to be 0.5.

Many papers since have sought to broaden the class of functions that can make exponential use of depth. Yarotsky (2017) and Liang & Srikant (2016) construct $y = x^2$ on $[0, 1]$ with exponential accuracy by using triangle waves. To produce their approximation, start with $y = x$, then subtract $T(x)/4$, followed by $T(T(x))/16$, etc... Figure 1 graphs the convergence of the approximation to $x^2$ and the convergence of their derivatives. This approximation technique intuitively feels very fundamental to how ReLU networks operate, so we would like to use modified triangle waves to generalize it, seeing if it can represent a greater array of one-dimensional convex curves. It is particularly interesting that the derivative of the $y = x^2$ approximation approaches $y = 2x$ in the infinite depth limit. This is a sufficient, but not necessary property for convergence to differentiable functions. We place a particular emphasis on investigating whether this property is important for ReLU networks when approximating differentiable functions.

Generalizing the $x^2$ approximation is important because it is widely used by other theoretical works as a building block to guarantee exponential convergence rates in more complex systems. One possible use case is to construct a multiplication gate. Perekrestenko et al. (2018) does so via the identity $(x + y)^2 = x^2 + y^2 + 2xy$. The squared terms can all be moved to one side, expressing the product as a linear combination of squared terms. They then assemble these gates into an interpolating polynomial, which can have an exponentially decreasing error when the interpolation points are chosen to be the Chebychev nodes. But polynomial interpolation does not scale well into high dimensions, so this and papers with similar approaches will usually come with restrictions that limit function complexity: Wang et al. (2018) requires low input dimension, Montanelli et al. (2020) uses band limiting, and Chen et al. (2019) approximates low dimensional manifolds. With a generalization of the $x^2$ operation, it might be possible to achieve an exponentially decaying error with depth in one dimension that does not rely on polynomial interpolation. This additional flexibility might remove the bottleneck into higher dimensions in future works.

Much work is also focused on showing how ReLU networks can encode and subsequently surpass traditional approximation methods (Lu et al., 2021), (Daubechies et al., 2022). Interestingly, some of the themes from above like composition, triangles, or squaring are still present. One other interesting comparison is to (Ivanova & Kubat, 1995) which uses decision trees as a means to initialize neural networks. It is a sigmoid/classification analogy to this work, but rather than an attempting to improve neural networks with decision trees, it is an attempt to improve decision trees with neural networks.

### 2.2 RELATED WORK IN NETWORK INITIALIZATION

Our work seeks to improve network initialization by making use of explicit theoretical results. This stands in sharp contrast the current standard approach. Two popular initialization methods implemented in PyTorch are the Kaiming (He et al., 2015) and Xavier initialization (Glorot & Bengio, 2010). They use weight values that are sampled from distributions defined by the input and output dimension of each layer. Aside from suboptimal approximation power associated with random weights, a common issue is that an entire ReLU network can collapse into outputting a constant value. This is referred to as the dying ReLU phenomenon. It occurs when the initial weights and biases cause every neuron in a particular layer to output a negative value. The ReLU activation then sets the output of that layer to 0, blocking any gradient updates. As depth goes to infinity, this becomes increasingly likely (Lu, 2020). Several papers propose solutions: Shin & Karniadakis (2020) uses a data-dependent initialization, while Singh & Sreejith (2021) introduces an alternate weight distribution called RAAI that can reduce the likelihood of the issue and increase training speed. We observed during our experiments that this method greatly reduces, but does not eliminate the likelihood of dying ReLU. Our approach on the other hand is only minimally probabilistic, and the imposed weight structure prevents collapse in this manner.

## 3 COMPOSITIONAL NETWORKS

Here we discuss how to deliberately architect the weights of a 4 neuron wide ReLU network to induce an exponential number of linear segments. Throughout the paper we refer to these as compositional networks.

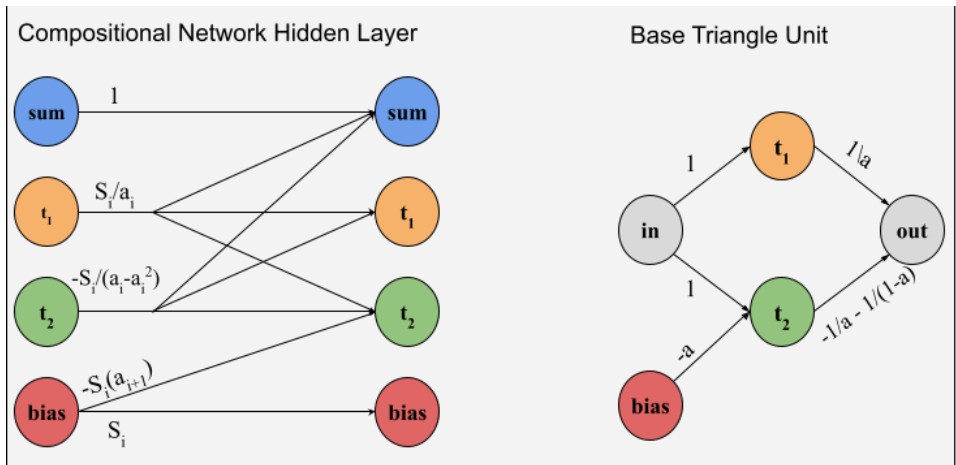

Figure 2: Hidden Layers of Compositional Networks

The subnetwork on the right generates a triangular shaped output. Its maximum output is 1 at the peak location $a \in (0, 1)$. Neuron $t_1$ simply preserves the input signal. Meanwhile $t_2$ is negatively biased, deactivating it for inputs less than $a$. Subtracting $t_2$ from $t_1$ changes the slope at the point where $t_2$ re-activates. The weight $-1/(a - a^2) = -(1/a + 1/(1 - a))$ is picked to completely negate $t_1$'s influence, and then produce a negative slope. When these components are stacked, the individual triangles they form will be composed. It can be beneficial in this setting to think about neurons in terms of their output over the entire input domain $[0, 1]$. $t_1$ neurons hold triangle waves, $t_2$ neurons hold an alternating sequence of triangles and inactive regions. This can be seen in Figure

3. Note how $t_1$ and $t_2$ have identical slopes.

The hidden layers in a compositional network consist of four neurons. Two of which are dedicated to composing triangle waves. Naively stacking the triangle units from figure 2 together would form a $1 \times 2 \times 1 \times 2 \times 1...$ shape. Instead, any outgoing weight from $t_1$ or $t_2$ is shared, every neuron taking in a triangle wave does so by combining $t_1$ and $t_2$ in the same proportion. This replicates a one dimensional output without the use of extra neurons. The accumulation neuron passes a weighted sum of all previous triangle waves through each layer. If this were naively implemented, it would multiply the $t_1$ and $t_2$ weights by the sum coefficients. These coefficients are exponentially decaying, so learning these weights directly may cause

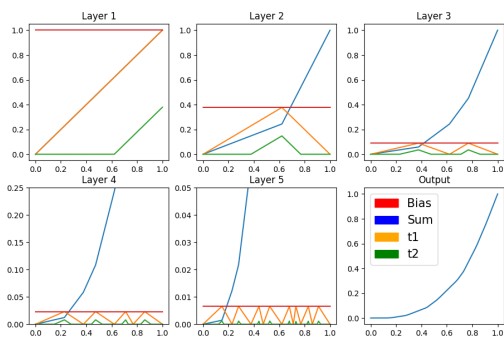

Figure 3: Individual neuron outputs with respect to network input

conditioning issues. Instead, the ratios between the coefficients are distributed amongst all the weights, so that the outputs of neurons 2 and 3 decay in amplitude in each layer. A conventional bias will have no connections to prior layers, so it will be unable to adjust to the weight decay. Therefore a fourth neuron is configured to output a constant signal. Other neurons can then use their connection to it as a bias. This neuron will then connect to itself so that it can scale down with each layer.

The network structure described above is the backbone of our training manifold. Selecting weights in this manner always creates an exponential number of bends in the model output. By training the triangle peaks, and the sum coefficients, we can traverse parameter space while maintaining maximal ReLU usage. One potential issue with this approach is that summing self-composed triangle waves with arbitrary scales will most likely produce a fractal curve, which will not usually resemble a differentiable function. We suspect that their inclusion in the training manifold can lead to many local minima. To address this, we experiment with enforcing that the network output be differentiable (were the network extended infinitely). We find that enforcing this additional constraint helps further steer the training process away from local minima, significantly improving mean error values.

### 3.1 REPRESENTING DIFFERENTIABLE FUNCTIONS

In this section we derive a necessary condition for a differentiable model output in the infinite depth limit. We show that the peak parameters entirely determine the appropriate sum coefficients. In the appendix we show that this parameter choice is also sufficient if the peak parameters do not approach 0 or 1. To begin we define a triangle function as

$$T_i(x) = \begin{cases} \frac{x}{a_i} & 0 \le x \le a_i \\ 1 - \frac{x-a_i}{1-a_i} & a_i \le x \le 1 \end{cases}$$

where $0 \le a_i \le 1$. This produces a triangular shape with a peak at $x = a_i$ and both endpoints at $y = 0$. Each layer of the network would compute these if directly fed the input signal. Its derivatives are the piecewise linear functions:

$$T_i'(x) = \begin{cases} \frac{1}{a_i} & 0 < x < a_i \\ \frac{1}{1-a_i} & a_i < x < 1 \end{cases} \tag{1}$$

In the actual network, the layers feed into each other, composing the triangle waves:

$$W_i(x) = \bigcirc_{j=0}^{i} T_j(x) = T_i(T_{i-1}(...T_0(x))) \tag{2}$$

The primary function of interest represents the output of one of our networks, were it to be infinitely deep:

$$F(x) = \sum_{i=0}^{\infty} s_i W_i(x) \tag{3}$$

where $s_i$ are scaling coefficients on each of the composed triangular waveforms $W_i$. We would like to select the $s_i$ based on $a_i$ in a manner where the derivative $F'(x)$ is defined on all of $[0, 1]$ which can only be done if the left and right hand derivative limits $F'_+(x)$ and $F'_-(x)$ agree.

Notationally we will denote the sorted x-locations of the peaks and valleys of $W(x)$ by the lists $P_i = \{x : W_i(x) = 1\}$ and $V_i = \{x : W_i(x) = 0\}$. We will use the list $B_i$ to reference the locations of all non-differentiable points, which we refer to as bends. $f_i(x) = \sum_{n=0}^{i-1} s_n W_n(x)$ will denote finite depth approximations up to but not including layer $i$. The error function $E_i(x) = \sum_{n=i}^{\infty} s_n W_n(x) = F(x) - f_i(x)$ to represent the error between the finite approximation and the infinite depth network.

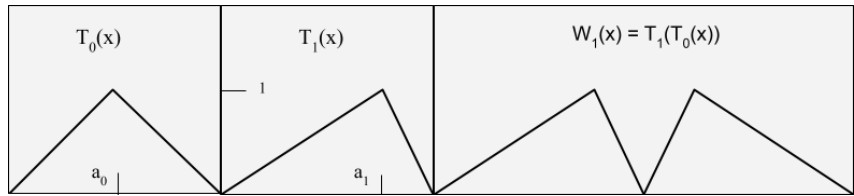

Figure 4: Triangle functions and composed triangle wave

Figure 4 highlights some important properties (proved later) about composing triangle functions. Peaks alternate with valleys. Peak locations in one layer become valleys in the next, and valleys persist. To produce $W_i$, each line segment of $W_{i-1}$ becomes a dilated copy of $T_i$. On downward slopes, the input to layer $i$ is reversed, so those copies are backwards. Each triangle function has two distinct slopes $1/a_i$ and $-1/(1 - a_i)$ which are dilated by the chain rule during the composition. Because the triangles are situated back to back, the slopes of $W_i$ on each side of a valley are proportional. New valleys are scaled by $1/(1 - a_i)$, while old ones are scaled by $1/a_i$.

Next we compute the derivative of the error function $E'_i(x)$ at points in $P_i$. Notice that the finite approximation $f_i(x)$ is differentiable at $x$ because it excludes layer $i$ by definition. The addition of $W_i$ is the first time a "gap" opens up at $F'(x)$, we would like to reason about the outstanding terms in the sum so that in the limit continuity may be restored.

**Lemma 3.1.** *For any point $x \in P_i$, $E'_+(x)$ and $E'_-(x)$ are proportional to*

$$s_i - \frac{1}{1 - a_{i+1}}\left(s_{i+1} + \sum_{n=i+2}^{\infty} s_n \prod_{k=i+2}^{n} \frac{1}{a_k}\right). \tag{4}$$

*Proof.* Let $x_n$ be some point in $P_i$, and let $n$ be its index in any list it appears in. To calculate the value of $E'_+(x)$ and $E'_-(x)$, we will have to find the slope of the linear intervals to the immediate left and right of $x_n$ for all $W_i$. The terms in the summation mostly derive from the chain rule.

We will denote $W'_{i+}(x)$ and $W'_{i-}(x)$ as $R_x$ and $L_x$. The first term of each sum will be $R_x s_i$ or $L_x s_i$. These are the sizes of the initial discontinuities. The second terms in the sum will be $-R_x \frac{s_{i+1}}{(1-a_{i+1})}$ and $-L_x \frac{s_{i+1}}{(1-a_{i+1})}$. We arrive at these by the chain rule, $W'_{i+1}(x) = T'_{i+1}(W_i(x))W'_i(x)$. There are 2 different possible slope values of $T_{i+1}$, the correct one to use is $-1/(1 - a_{i+1})$ because $x_n$ is a peak of $W_i$, so $W_i(x) > a_{i+1}$ for $x \in (B_{i+1}[n-1], B_{i+1}[n+1])$. Note that the second terms have the opposite sign as the first, shrinking the discontinuities.

For all remaining terms, since $x_n$ was in $P_i$ it is in $V_j$ for $j > i$. For $x \in (B_{j+1}[n-1], B_{j+1}[n+1])$, $W_j(x) < a_{j+1}$ and the chain rule applies the first slope $1/a_{j+1}$. Since this slope is positive, every term has the opposite sign as the first, and the discontinuities close monotonically. Summing up all the terms with the coefficients $s_i$, and factoring out $R_x$ and $L_x$ will yield the formula. Since these initial slopes can factor out of all terms, this equation proportionally describes the behavior of $E'(x)$ for all $x \in P_i$. □

**Lemma 3.2.** *If $E'(x)$ is defined it must be equal to 0*

*Proof.* Let $S$ represent equation 4, and $R_x$ and $L_x$ be the constants of proportionality. If $E'_+ = E'_-$, then $R_x S = L_x S$ for all $x \in P_i$. Since $W_i$ is made of oscillating line segments, $R_x$ and $L_x$ have opposite signs, and so the only way to satisfy the equation is if $S = 0$. Consequently, $E'(x) = 0$ for all $x \in P_i$. □

This lemma shows that the derivative of the finite approximation excluding $W_i$ is the same as that of the infinite sum.

**Lemma 3.3.** *for all $x \in P_i$:*

$$F'(x) = f_i'(x) = \sum_{j=0}^{i-1} s_j W_j'(x) \tag{5}$$

*Proof.* From the previous lemma we know $E'(x) = 0$. $F(x) = \sum_{j=0}^{i-1} s_j W_j(x) + E(x)$. The sum of the first $i - 1$ terms is differentiable at the points $P_i$ since they lie between the discontinuities at $B_{i-1}$. □

Next we prove our main theorem, the idea is that much of the formula for $E'(x)$ will be shared between two successive generations of peaks. Once they are both valleys they will behave the same, so the sizes of their remaining discontinuities will need to be proportional.

**Theorem 3.4.** *$F'(x)$ is continuous only if the scaling coefficients are selected based on $a_i$ according to:*

$$s_{i+1} = s_i(1 - a_{i+1})a_{i+2} \tag{6}$$

*Proof.* By rewriting equation equation 4 for layers $i$ and $i + 1$ in the following way:

$$s_i(1 - a_{i+1}) = s_{i+1} + \frac{1}{a_{i+2}}\left(s_{i+2} + \sum_{n=i+3}^{\infty} s_n \prod_{k=i+3}^{n} \frac{1}{a_k}\right)$$

$$s_{i+1}(1 - a_{i+2}) = s_{i+2} + \sum_{n=i+3}^{\infty} s_n \prod_{k=i+3}^{n} \frac{1}{a_k}$$

allows for a substitution to eliminate the infinite sum

$$s_i(1 - a_{i+1}) = s_{i+1} + \frac{1 - a_{i+2}}{a_{i+2}} s_{i+1}$$

collecting all the terms gives

$$s_{i+1} = \frac{s_i(1 - a_{i+1})}{1 + \frac{1 - a_{i+2}}{a_{i+2}}}$$

which simplifies to the desired result. □

## 4 Experiments

In this section we show how the usage of training manifolds may produce better approximations to convex functions. Our principal technique takes place in two stages. First, gradient descent is applied to the parameters which define a compositional network. This produces a better loss landscape and forces the retention of bends while approaching the target function. Secondly, standard gradient descent is then used to fine tune the underlying matrix values.

In this section we highlight several important comparisons. To demonstrate the benefits of the reparameterized loss landscape, we compare our approach against networks that were initialized onto the differentiable manifold, but then immediately had their raw weights trained. We benchmark against PyTorch's default settings (nn.linear() uses Kaiming initialization) as well as the RAAI distribution from Singh & Sreejith (2021) and produce errors that are orders of magnitude lower than both. Lastly we compare the differentiable manifold to the manifold where the scaling parameters are free showing the benefits of differentiability constraints.

### 4.1 Experimental Setup

All models are trained using ADAM Kingma & Ba (2017) as the optimizer for 1000 epochs to ensure convergence. The data are $2^9$ evenly spaced points on the interval $[0, 1]$ for each of the curves. Each network is four neurons wide with five hidden layers, along with a 1-dimensional input and output. The loss function used is the mean squared error, the average and minimum loss are recorded for 30 models of each type. The four curves we approximate are $x^3$, $x^{11}$, $\tanh(3x)$, and a quarter period of a sine wave. To approximate the sine and the hyperbolic tangent, the triangle waves are added to the line $y = x$. For the other approximations, the waves are subtracted. This requires the first scaling factor to be $a_0 * a_1$ instead of $(1 - a_0) * a_1$.

Although we previously discuss two stages, we technically divide training into three stages to compare the two possible manifolds. Stage 1 is optimization over the manifold of peak location parameters $a$, the scaling coefficients $s$ are constrained such that the model output is differentiable in the infinite depth limit. Stage 2 allows the scales to decouple from the peaks, but enforces exponentially many bends. Stage 3 performs gradient descent over the entire set of raw network parameters. It is possible to transition between all three stages by successively relaxing constraints. A stage can be bypassed by relaxing constraints without having performed the associated optimization. By circumventing training steps, it allows the relative importance of each stage to be assessed. For example " Stage 1 and 3" in the tables is training on the differentiable manifold, skipping the intermediate manifold, and then doing regular gradient descent. All initial peak parameters are selected from the same random seed, so these comparisons use a common set of starting points. The models labeled "Free Scale Initialization" deviate slightly from the pattern, they initialize onto the non-differentiable manifold, and use random values between 0 and 1 as the scales. These are not deterministically related to the random seed.

### 4.2 Numerical Results

There are two main trends that emerge in the data. First, the worst performing networks are those that rely on purely randomized initializations. The act of initializing onto a manifold alone is enough to make an improvement (shown by stage 3). Training on a manifold leads to a reduction in minimum error by three orders of magnitude. The second observation is that using the differentiable manifold not only leads to the highest performing models, but it also closes the gap between the minimum and mean errors. This indicates that these loss landscapes are indeed the most reliable. When stage 1 is bypassed the min-mean gap rises considerably and the minimum error roughly doubles. Little changes when stage 2 is bypassed. Doing so can sometimes offer improvements over using all three training stages. Around half of the default networks collapse, which is a big driver of the poor average performance. RAAI is able to eliminate most, but not all of the dying ReLU's due to its probabilistic nature. These skew its mean error. By skipping the differentiable manifold in stage

1, the "stage 2 and 3" models find a very suboptimal solution on $x^3$ and $x^{11}$ for a particular input that further drags down the average. If training is halted after using the differentiable manifold, there remains a sizable error reduction which can be achieved by gradient descent. This suggests that this choice of training manifold is not highly expressive, containing a rather limited selection of functions.

Table 1: MSE error approximating $y = x^3$ and $x^{11}$

| Network | Min $x^3$ | Min $x^{11}$ | Mean $x^3$ | Mean $x^{11}$ |
|---|---|---|---|---|
| Default Network | $2.11 \times 10^{-5}$ | $2.19 \times 10^{-5}$ | $7.20 \times 10^{-2}$ | $2.82 \times 10^{-2}$ |
| RAAI Distribution | $2.14 \times 10^{-5}$ | $4.40 \times 10^{-5}$ | $3.97 \times 10^{-2}$ | $4.12 \times 10^{-2}$ |
| Stage 1 | $3.41 \times 10^{-5}$ | $3.31 \times 10^{-4}$ | $4.32 \times 10^{-4}$ | $2.65 \times 10^{-3}$ |
| Stage 3 (GD only) | $3.42 \times 10^{-6}$ | $1.86 \times 10^{-5}$ | $1.68 \times 10^{-4}$ | $3.56 \times 10^{-4}$ |
| Free Scale Init. | $2.55 \times 10^{-7}$ | $\mathbf{6.82 \times 10^{-7}}$ | $6.81 \times 10^{-5}$ | $1.38 \times 10^{-3}$ |
| Stage 2 and 3 | $1.64 \times 10^{-7}$ | $3.20 \times 10^{-6}$ | $2.57 \times 10^{-2}$ | $2.83 \times 10^{-1}$ |
| Stage 1 and 3 | $\mathbf{7.86 \times 10^{-8}}$ | $9.58 \times 10^{-7}$ | $\mathbf{6.57 \times 10^{-7}}$ | $5.30 \times 10^{-5}$ |
| All Stages | $1.87 \times 10^{-7}$ | $7.43 \times 10^{-7}$ | $1.62 \times 10^{-6}$ | $\mathbf{1.42 \times 10^{-5}}$ |

Table 2: MSE error approximating $y = \sin(x)$ and $y = \tanh(3x)$

| Network | Min $\sin(x)$ | Min $\tanh(3x)$ | Mean $\sin(x)$ | Mean $\tanh(3x)$ |
|---|---|---|---|---|
| Default Network | $4.50 \times 10^{-5}$ | $5.75 \times 10^{-5}$ | $1.15 \times 10^{-1}$ | $1.96 \times 10^{-1}$ |
| RAAI Distribution | $3.59 \times 10^{-5}$ | $1.09 \times 10^{-5}$ | $3.63 \times 10^{-2}$ | $2.31 \times 10^{-2}$ |
| Stage 1 | $6.00 \times 10^{-6}$ | $2.83 \times 10^{-5}$ | $9.73 \times 10^{-5}$ | $3.48 \times 10^{-4}$ |
| Stage 3 (GD only) | $3.75 \times 10^{-7}$ | $1.07 \times 10^{-6}$ | $1.93 \times 10^{-5}$ | $8.38 \times 10^{-5}$ |
| Free Scale Init. | $1.29 \times 10^{-7}$ | $1.17 \times 10^{-7}$ | $1.23 \times 10^{-4}$ | $4.67 \times 10^{-4}$ |
| Stage 2 and 3 | $6.12 \times 10^{-8}$ | $1.47 \times 10^{-7}$ | $9.26 \times 10^{-6}$ | $1.56 \times 10^{-4}$ |
| Stage 1 and 3 | $5.06 \times 10^{-8}$ | $\mathbf{6.82 \times 10^{-8}}$ | $3.14 \times 10^{-7}$ | $9.58 \times 10^{-7}$ |
| All Stages | $\mathbf{3.98 \times 10^{-8}}$ | $1.04 \times 10^{-7}$ | $\mathbf{1.04 \times 10^{-7}}$ | $\mathbf{2.45 \times 10^{-7}}$ |

## 4.3 INSIDE THE COMPOSITIONAL NETWORKS

Figure 3 (used earlier) further confirms our suspicion that $x^3$ is not in the differentiable manifold. A network trained solely on the manifold generates exponentially many bends but cannot place them freely in optimal locations. This leads to a somewhat wobbly appearance.

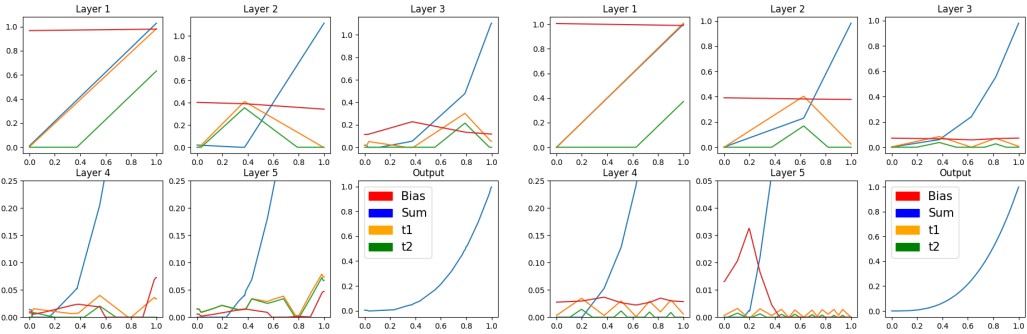

Figure 5: Training on differentiable manifold first retains better structure

In figure 5 we compare initialization on the differentiable manifold followed by gradient descent against training on the manifold first. We observe that without the guidance of the manifold,

gradient descent usually loses the triangle generating structure around layer 4 or 5, devolving into noisy patterns. By first using the manifold, effective ReLU usage can be extended to greater depths (we observed around 6 or 7 layers). Theoretical works often rely on specific constructions within networks to prove results, but here we observe that gradient descent readily abandons any such structure in favor of worse models. A theoretical result without a subsequent plan to control training will likely not succeed. Part of what explains the success of our approach is that we use unconstrained gradient descent for much less of our traversal of parameter space. If we were to substitute our training manifold for a more expressive one, it could lessen or possibly eliminate the gap between the manifold and the target, leading to deeper bend retention.

## 4.4 INSIDE THE DEFAULT NETWORK

Figure 6 shows the interior of A default network, unlike in the previous figures, the layers here are shown before applying ReLU. The default networks fail to make efficient use of ReLU to produce bends, even falling short of 1 bend per neuron, which can easily be attained by forming a linear spline (1 hidden layer) that interpolates some of the data points. Rather than an exponential efficiency boost, depth is actually hindering these networks. Examining the figure, the first two layers are wasted. No neuron's activation pattern crosses $y = 0$, so ReLU is never used. Layer 3 could be formed directly from the input signal. Deeper in the network, more neurons remain either strictly

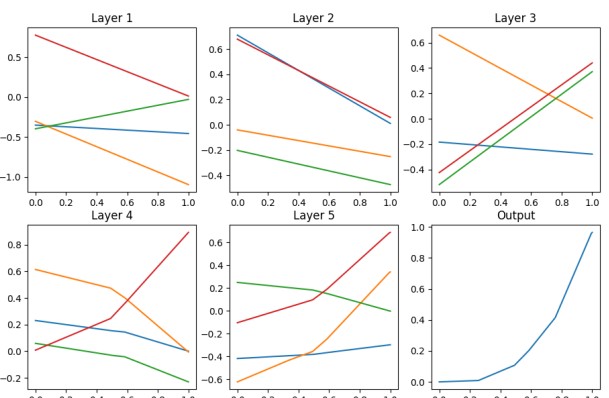

Figure 6: Approximation produced by a default Network

positive or negative. Those that intersect $y = 0$ are monotonic, only able to introduce one bend at a time. The core issue is that while more bends leads to better accuracy, networks that have few bends are not locally connected in parameter space to those that have many. This is problematic since gradients can only carry information about the effects of infinitesimal modifications to the slope and bias. For a neuron that always outputs a strictly positive value (such as the red in layer 2), bends cannot be introduced by infinitesimal adjustments. Therefore bend-related information will be absent from its gradients. If a network ever learns to produce a new bend, it is due to indirectly related local factors that have incentivized neurons to produce negative outputs.

## 5 EXTENSIONS

Although this work investigates the simplest class of problems imaginable, there is a great deal of complexity involved in reasoning about individual neurons in a network and their usage. Whenever insights into neuron structure become available in higher dimensional problems, this work provides a blueprint for how to translate those insights into an accompanying training procedure. The success of our method may indicate that there is a nearby theoretical model which can exponentially converge to convex functions. With that network, it may be possible to assemble new approximation schemes. Therefore it is of great theoretical importance to determine whether the eventual loss of bends with depth is due only to the weakness of gradient descent, or if exponential error decay fundamentally cannot continue to infinite depth with an architecture similar to this. Lastly, the question of enforcing differentiability is very interesting. We show here that it is a great way to constrain against "bad" fractals, but that it limits possible representations. Furthermore, we show in the appendix that a network with this architecture with a defined second derivative necessarily approximates $x^2$. This suggests that the approximation power of our networks may be coming from fractal representations near the differentiable models in parameter space.

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

# A APPENDIX

## A.1 SUFFICIENCY FOR DIFFERENTIABILITY

We can show that in addition to being necessary for continuity of the derivative, our choice of scaling is sufficient when $a_i$ are bounded away from 0 or 1

**Theorem A.1.** $F'(x)$ *is continuous if we can find $c > 0$ such that $c \leq a_i \leq 1 - c$ for all $i$*

*Proof.* We begin by considering equation 4 for layer $i$.

$$s_i = \frac{1}{1 - a_{i+1}}(s_{i+1} + \sum_{n=i+2}^{\infty} s_n \prod_{k=i+2}^{n} \frac{1}{a_k})$$

Recall that this equation is telling us about the size of the discontinuities in the derivative as triangle waves are added. Each time a triangle wave is added, it can be thought of as subtracting the terms on the right. We will prove our result by substituting equation 6 into this formula, and then verifying that the resulting equation is valid. First we would like to rewrite each $s_n$ in terms of $s_i$. equation 6 gives a recurrence relation. Converting it to an explicit representation we have:

$$s_n = s_i(\prod_{j=i+1}^{n} 1 - a_j)(\prod_{k=i+2}^{n+1} a_k) \tag{7}$$

When we substitute this into equation 4, three things happen: each term is divisible by $s_i$ so $s_i$ cancels out, every factor in the product except the last cancels, and $1 - a_{i+1}$ cancels. This leaves

$$1 = a_{i+2} + (1 - a_{i+2})a_{i+3} + (1 - a_{i+2})(1 - a_{i+3})a_{i+4} + ... = \sum_{n=i+2}^{\infty} a_n \prod_{m=i+2}^{n-1}(1 - a_m) \tag{8}$$

We will now argue that each term of the sum on the right accounts for a fraction (equal to $a_i$) of the remaining error. Inductively we can show:

$$1 - \sum_{n=i+2}^{j} a_n \prod_{m=i+2}^{n-1} (1 - a_m) = \prod_{m=i+2}^{j} (1 - a_m) \tag{9}$$

intuitively this means as long as the first term appearing on the right is repeatedly subtracted, that term is always equal to $a_n$ times the left side. As a base case we have $(1 - a_{i+2}) = (1 - a_{i+2})$. Assuming the above equation holds for all previous values of $j$

$$1 - \sum_{n=i+2}^{j+1} a_n \prod_{m=i+2}^{n-1} (1 - a_m)) = 1 - \sum_{n=i+2}^{j} a_n \prod_{m=i+2}^{n-1} (1 - a_m)) - a_{j+1} \prod_{m=i+2}^{j} (1 - a_m)) =$$

using the inductive hypothesis to make the substitution

$$\prod_{m=i+2}^{j} (1 - a_m)) - a_{j+1} \prod_{m=i+2}^{j} (1 - a_m)) = \prod_{m=i+2}^{j+1} (1 - a_m))$$

Since all $c < a_i < 1 - c$, the size of the discontinuity at the points $P_i$ is upper bounded and lower bounded by exponentially decaying series. Since both series approach zero, so does the series here. $\square$

## A.2  Second Derivatives

Here we show that any function represented by one of these networks that is not $y = x^2$ does not have a continuous second derivative. To show this we will sample a discrete series of $\Delta y / \Delta x$ values from $F'(x)$ and show that the left and right hand limits are not equal (unless $a_i = 0.5$), which will imply that the continuous version of the limit for the second derivative does not exist. First we will produce the series of $\Delta x$. Let $x$ be the location of a peak at layer $i$, and let $l_n$ and $r_n$ be its immediate neighbors in $B_{i+n}$.

**Lemma A.2.** *If $c < a_i < 1 - c$ for all $i$, we have $\lim_{n \to \infty} r_n = \lim_{n \to \infty} r_n = x$. Furthermore, $r_n, l_n \neq x_n$ for any finite $i$.*

*Proof.* Let $K$ be a constant of proportionality for the slopes on the left and right of $x$ in layer $i$. These slopes are proportional to $1/a_i$ and $1/(1 - a_i)$ depending on which way the triangle encompassing $x$ is oriented ($W'_{i-1}$ could have been positive or negative at its location), We will denote these as $1/L$ and $1/R$ and reason about them later. $x_n$ is a peak location of $W_i$, so on the left side slope is negative and the right is positive. Solving for the location of $T_{i+1}(W_i(x)) = 1$ on each side will give $l_1 = x - (1 - a_{i+1})L/K$ and $r_1 = x + (1 - a_{i+1})R/K$.

On each subsequent iteration $i + n$, $x$ is a valley point and the $\Delta x$ intervals get multiplied by $a_{i+n}$. $x$ is a valley point so the left slope is positive and the right is negative, and $l_n, r_n$ are peak points. The slope magnitudes are given by $\frac{1}{x - l_n}$ and $\frac{1}{r_n - x}$ since $C_{i+n}$ oscillates from 0 to 1 over these spans. Solving for the new peaks again will give $l_{n+1} = x - a_{i+1}(x - l_n)$ and $r_{n+1} = x + a_{i+1}(r_n - x)$. The resulting non-recursive formulas are:

$$x - l_n = \frac{L}{K}(1 - a_{i+1}) \prod_{m=2}^{n} a_{i+m} \text{ and } r_n - x = \frac{R}{K}(1 - a_{i+1}) \prod_{m=2}^{n} a_{i+m} \tag{10}$$

The right hand sides will never be equal to zero with a finite number of terms since $a$ parameters are bounded away from 0 and 1 by $c$. $\square$

Next we derive the values of $\Delta y$ to complete the proof.

**Theorem A.3.** *A compositional network with a second differentiable output necessarily outputs $y = x^2$*

*Proof.* The points $l_n$ and $r_n$ are all peak locations, equation 5 gives their derivative values as $f'_{i+n}(r_n)$. Earlier we reasoned about the sizes of the discontinuities in $F'(x)$ at $x$, since $l_n$ and $r_n$ always lie on the linear intervals surrounding $x$ as $n \to \infty$, we can get the value of $f'_i(x) - f'_{i+n}(r_n)$ using equation 9 with the initial discontinuity size set to $s_i(K/R)$ rather than 1. Focusing on the right hand side we get:

$$f'_i(x) - f'_{i+n}(r_n) = s_i(K/R) \prod_{m=2}^{n} (1 - a_{i+m})$$

taking $\Delta y / \Delta x$ gives a series:

$$\frac{K^2 s_i}{R^2(1 - a_{i+1})} \prod_{m=i+2}^{n} \frac{1 - a_m}{a_m}$$

The issue which arises is that the derivation on the left is identical, except for a replacement of $R$ by $L$. The only way for these formulas to agree then is for $R = L$ which implies $a_i = 1 - a_i = 0.5$ $\square$

### A.3 ERROR DECAY

Lastly we show that the error of these approximations decays exponentially

**Lemma A.4.** *The ratio $s_{i+2}/s_i$ is at most* $0.25$

*Proof.* by applying formula equation 6 twice, we have

$$s_{i+2} = s_i(1 - a_{i+1})(1 - a_i + 2)a_{i+2}a_{i+3}$$

To maximize $s_{i+2}$ we choose $a_{i+1} = 0$ and $a_{i+3} = 1$. The quantity $a_{i+2} - a_{i+2}^2$ is a parabola with a maximum of $0.25$ at $a_{i+2} = 0.5$ $\square$

**Lemma A.5.** *The function $F(x)$ is convex*

*Proof.* To establish this we will introduce the list $Y'_i = [F'(V_i[0]), f'_i(V_i[n]), F'(V_i[2^i])]$, which tracks the values of the derivative at the $i$th set of valley points. All but the first and last points will have been peaks at some point in their history, so equation 5 gives the value of those derivatives as $f'_i$.

We establish an inductive invariant that the y-values in the list $Y_i$ remain sorted in descending order, and that $Y'_i[n] \ge f'_i(x) \ge Y'_i[n+1]$ for $V_i[n] < x < V_i[n+1]$.

Before any of $W_i$ are added, $f_0$ is a line with derivative 0, $V_0$ is its two endpoints. $Y'_0$ is positive for the left endpoint (negative for right) since on the far edges $F'$ is a sum of a series of positive (or negative) slopes, Therefore the points in $Y'$ are in descending sorted order. The second part of the invariant is true since 0 is in between those values.

Consider an arbitrary interval $(V_i[n], V_i[n+1])$ of $f_i$, this entire interval is between two valley points, so $f'_i$ (which hasn't added $W_i$ yet) is some constant value in between $Y'_i[n]$ and $Y'_i[n+1]$. The point $x \in P_i \cap (V_i[n], V_i[n+1])$ will have $F'(x) = f'_i(x)$, and it will become a member of $V_{i+1}$. This means we will have $Y_{i+1}[2n] > Y_{i+1}[2n+1] > Y_{i+1}[2n+2]$, maintaining sorted order.

Adding $s_i W_i$ takes $f_i$ to $f_i + 1$ splitting each constant valued interval in two about the points $P_i$, increasing the left side, and decreasing the right side. Recalling from the derivation of equation 4 all terms but the first in the sum have the same sign, so the values in $Y'_i$ are approached monotonically. Therefore on the left interval we have $Y'_i[n] > f_{i+1} > f_i$ and on the right we have $f_i > f_{i+1} > Y'_i[n+1]$. And so $f_{i+1}$ remains monotone decreasing. $\square$

**Theorem A.6.** *The approximation error $E_i(x)$ decays exponentially*

*Proof.* To get this result we apply the previous two lemmas. Since $F$ is convex, it lies above any line segment connecting any two points on the curve. $W_i(x) = 1$ for all $x \in P_i$, but $W_i(x) = 0$

for points in $V_i$. Since the bend points are only of nonzero value once, $f_i(x) = F(x)$ for all points in $V_i$. $f_i$ is made of line segments and equals $F$ repeatedly, $E_i$ will be a series of positively valued curve segments. The derivative will still be decreasing on each of these intervals since it was just shifted by a constant, and each of these intervals will be convex itself.

Since $E'(x) = 0$ for $x \in P_i$, these points will be the locations of maximum error. Since they only have nonzero value in $W_i$, and $E$ is convex, $s_i W_i(x) = max(E) = s_i$. Since $s_i$ decay exponentially we have our result. $\square$

