# OpenReview forum: "A trainable manifold for accurate approximation with ReLU Networks"
_ICLR.cc/2024/Conference — Submitted to ICLR 2024_

### Official Review · Reviewer_FQgq · 2023-10-22

**Soundness:** 2 fair
**Presentation:** 3 good
**Contribution:** 2 fair
**Rating:** 3
**Confidence:** 3

**Summary:**

This paper discusses the learnability of a simple neural network architecture. Specifically the problem of fitting a quadratic function is of interest. This paper proposes a compositional network architecture where each base component is composed by four simple linear or nonlinear activation functions, and that the neural network architecture is replication of the base component. The paper demonstrates its learnability, in other words the power of this architecture that approximates a quadratic function.

**Strengths:**

I think this paper is readable and intuitive. Although it discusses a simple model, it is astonishing that the application of a single model can fit a different function with fast approximation rate. The content is precise and self-consistent, and there are many figures and discussions that help readers go through the approximation process. The proof is mathematically correct.

**Weaknesses:**

I think it would be great to discuss this paper in a bigger picture, for example, how does the power of approximation of this model architecture compare with other kinds of neural networks, especially how does it present a tradeoff between learnability and simpleness, and why is this architecture of interest. Since today's neural networks are complicated and they perform well with a lot of reasons, while this unusual neural network is seldomly used, how is the proof in this paper share the light on the analysis of other types of neural network models, and how does it guide the selection of neural network architectures, training algorithms, and the simple complexities required to train a model without overfitting etc.? With the answers to the above questions, I think the importance of this paper is better presented.

**Questions:**

See above.

---

### Official Review · Reviewer_xLCZ · 2023-10-28

**Soundness:** 3 good
**Presentation:** 2 fair
**Contribution:** 1 poor
**Rating:** 3
**Confidence:** 4

**Summary:**

This paper presents a trainable manifold with ReLU networks for function approximations, by reparameterizing the ReLU networks. This work is built upon the previous work on constructing the weights of the ReLU networks such that the network can generate triangle waves (Telgarsky 2015) which can be utilized to show an exponential separation between deep and shallow ReLU networks. The trainable parameters of the networks are $a_i$ which control the center of the triangle and $s_i$ which is the coefficient of the depth-$i$ composition of the triangle waves. Hanin and Rolnick, 2019 shows that the number of expected number of linear regions in a randomly initialized ReLU network does not scale exponentially with depth. Thus, the benefit of using the reparameterization (the authors proposed in this work) is that the output will have an exponential number of line segments. The authors show by their experiments that their initialization is able to produce smaller MSE error.

**Strengths:**

I personally find this idea of initializing the network to be on a manifold with exponential number of linear region novel and interesting, which connects theory and practice.

**Weaknesses:**

As for the current manuscript, the optimization and generalization property of such initialization is not sufficiently explored. Right now, only synthetic experiments are provided on some simple setting. Based on those simple setting, it seems that we can optimize the manifold, however, how hard it is to optimize such manifold in the real world setting is worth further studying. Further, in practice, the goal of people training a network is to hope the network can generalize. It is worth further investigation on the generalization property of the network. Since the network output has exponentially many linear region (and thus more capacity to fit), one may suspect that the network is highly vulnerable to input noise such that network overfits the data and is not able to generalize.

**Questions:**

I find the second paragraph of section 4.1 confusing. In table 1, what is the difference between default network and stage 3 (GD only)? The default network is Kaiming initialized but the stage 3 network is initialized with exponential number of linear region?

**Details Of Ethics Concerns:**

None.

---

> ### Author Response · Authors · 2023-11-23
>
> You're definitely correct in your point about the overfitting and generalization. The reason most of the experiments here are focused on interpolating perfect data is that I originally approached this paper in light of some of the theoretical literature where their main concern is whether certain representations exist in the set of neural networks. But given that this paper is concerned with whether efficient representations are learnable, I should probably place a greater emphasis on this.
>
> In the experiments in this paper, some of the functions become very jagged if the scaling parameters are not controlled. This is related to the problem of overfitting, the difference is that they have settled into a local minimum that is ineffective on the training set. The differentiability constraint in training eliminates these instances (the starting locations are all the same). It seems to provide an amount of protection from this issue by producing a starting point for standard gradient descent that is biased towards smoothness. It could be interesting to see if that behavior carries over to other settings

---

### Official Review · Reviewer_jsTK · 2023-11-01

**Soundness:** 3 good
**Presentation:** 4 excellent
**Contribution:** 3 good
**Rating:** 8
**Confidence:** 4

**Summary:**

A ReLU network trained with gradient descent in the parameter space does not efficiently leverage the usage of the linear segments given by ReLUs. To address this issue, the paper proposes a new reparameterization such that the output of a ReLU network is guaranteed to output a sawtooth-like function with exponentially many more linear segments as depth is increased. Since this procedure also creates many more discontinuities on the optimization landscape, a differentiability constraint is added to the reparameterization to steer the solution away from bad local minima. Such a constraint is derived from analyzing the derivative of an infinite-layer ReLU network. A theorem about this differentiability constraint is proved. Computer simulations are also provided to evaluate the proposed reparameterization. Several nonlinear functions are tested, and the results are promising.

**Strengths:**

- Originality: Most existing approximation results rely on the base function $x^2$, however, the structure of this base function can be destroyed by gradient descent and then hinder the approximation performance. This paper provides a novel optimization approach that constrains the optimization on a low-dimensional manifold such that superior approximation performance can be achieved. This new method is new in the sense that the exponentially many linear segments of a sawtooth-like function are preserved during training. Furthermore, along with a novel differentiability constraint, bad local minima can be avoided.

- Quality and clarity: This paper gives a comprehensive presentation on the approximation results of ReLU networks based on $x^2$. Using that argument, the paper seamlessly leads the read to understand how exponentially many linear segments can be preserved by designing a low-dimensional manifold. The part where the authors limit the manifold further with a differentiability constraint is also well presented. The whole paper is fairly well-written and easy to follow. I very much enjoy reading the paper.

- Significance: This new low-dimensional manifold idea greatly improves the optimization accuracy of ReLU networks for nonlinear function approximation. The improvements are significant, and the results are well supported by the theoretical justification in the paper.

**Weaknesses:**

1. The proposed optimization approach can be more useful if the reparameterization argument can be extended to high dimensional problems. It would be clearer for the reader if the authors can describe the main difficulties of such extension.

2. The nonlinear functions picked in Section 4.2 seem to be simple. It would be more convincing if the authors could also show superior performance for complicated nonlinear functions. Is preserving the linear segments truly beneficial for learning general nonlinear functions? I think a discussion of the function family that is friendly to the proposed method is important.

**Questions:**

1. Is the proposed method sensitive to the selection of optimizer? Can SGD yield the same performance?

2. How many bits are used to represent the weights in the ReLU network?

3. The performance is reported in mean and min. How is the worst-case scenario? Perhaps adding the max metric in Appendix.

4. Please add a subtitle to Figure 5 to indicate which of them is using the differentiable manifold.

5. Section 2.1, there is a typo in “Since each layer converts …”

6. Section 3.1 page 5, where is the definition of W(x)?

7. Lemma 3.3 For all x…

8. Given the fractal nature of the sawtooth-like function, would the proposed method demonstrate superior performance on some fractal functions? For example, the Cantor function.

---

> ### Author Response · Authors · 2023-11-23
>
> I'm so glad you liked the paper! this was actually my first time submitting to a conference. It got buried in the appendix, but there's a proof that the differentiability constraint forces the function represented by the network to be convex. While this constraint is important for ensuring consistent performance, it also takes away a great deal of flexibility. This is part of why I had difficulty extending this method to more complex problems.
>
> Fully connected ReLU networks are great at representing fractals. If I remember correctly, the Perekrestenko et al. citation gives a novel approximation to the Weierstrass function that exponentially improves upon previously existing methods. I think it may be the case that the ability of neural networks to represent fractals could be leveraged to provide a more flexible replacement for the differentiability constraint in this paper. Perhaps non-convex functions require nowhere-differentiable representations, but somehow this can be controlled so as to prevent jaggedness in the infinite-depth limit.
>
> The iterated composition of triangle waves is also a textbook example (I just saw it in a textbook the other day in fact) of a chaotic system. The "spikes" in the limit are dense in the input domain, creating a sensitivity to small perturbations in the input. This is important because it allows the slopes of two nearby input points to be distinguished from each other. In a deep enough network, one will trigger a different neuron activation pattern. Two inputs producing the same activation pattern lie in a linear region of the represented function. And it would likely be undesirable to have linear regions show up in your infinitely deep representation.
>
> This is part of why leveraging depth exponentially in a neural network is a challenge. It requires a composition of chaotic features to be summed (possibly in a non-differentiable manner) that approaches nicely behaved functions in the limit.

---

### Official Review · Reviewer_PUUB · 2023-11-05

**Soundness:** 2 fair
**Presentation:** 2 fair
**Contribution:** 2 fair
**Rating:** 3
**Confidence:** 2

**Summary:**

This paper studies the approximation ability of ReLU networks by encoding complex operations into ReLU networks using smaller base components. The derivation in this paper can produce networks with exponentially many piecewise-linear segments. The author claims that Their construction can enable the training process to overcome drawbacks associated with random initialization. The authors conduct the experiments on some synthetic datasets.

**Strengths:**

- This paper is well organized and has clear illustration figures.

- This paper only requires four neurons per layer to approximate.

- According to Tables, the constructed neuron networks can achieve better performance when approximating the simple function such as $y=x^3$, $y = x^{11}$, $y = \sin(x)$, $y = \tanh(x)$.

**Weaknesses:**

- The presentation of this paper is not clear.

- It is unclear how to apply the technique to real applications.

- How to understand Theorem 3.4?

**Questions:**

- The author claims that their results are minimally probabilistic and thus can prevent weight collapse. Why? How to understand this claim, and which theorem supports it? The authors may better add some comments or remarks about that.

- The authors propose a new architecture, but how to initialize it, can it be potentially extended to other structures like CNN or transformer?

---

### Author Response · Authors · 2023-11-23
**Thank you**

I'd like to thank all the reviewers for their efforts. The commentary is very constructive and I will keep it in mind when making improvements on the paper. I had wanted to extend this methodology to more practical problems during this rebuttal period, but it seems that more theoretical development will be necessary to enhance the flexibility of this technique first.

---

### Meta-Review · Area_Chair_xxw5 · 2023-12-05

**Metareview:**

A work investigating an initialization method aimed at overcoming several drawbacks associated with existing ones (e.g., Xavier and Kaiming), such as dying ReLU units. The method proceeds by restricting to a manifold of weights so that the output of networks so defined utilizes exponentially many piecewise-linear segments.

The reviewers found the idea of initializing the network on a manifold promoting an exponential number of linear region along the training process novel and interesting, and generally appreciated the clear writing. However, all coincided that the examples provided in the work may be overly simplistic and that more evidence is needed to demonstrate wider applicability. The authors are encouraged to revise accordingly and resubmit.

**Justification For Why Not Higher Score:**

More empirical work is needed.

**Justification For Why Not Lower Score:**

N\A

---

### Decision · Program_Chairs · 2024-01-16

Reject